# Ultrasonographic Diagnosis and Computed Tomographic Confirmation of a Scapular Body Stress Fracture in an Elite Boxer: A Case Report

**DOI:** 10.3390/diagnostics15202565

**Published:** 2025-10-11

**Authors:** Yonghyun Yoon, King Hei Stanley Lam, Jihyo Hwang, Seonghwan Kim, Jangkeun Kye, Hyeeun Kim, Junhan Kang, Jaeyoung Lee, Daniel Chiung-Jui Su, Teinny Suryadi, Anwar Suhaimi, Kenneth Dean Reeves

**Affiliations:** 1Department of Orthopedic Surgery, Hallym University Gangnam Sacred Heart Hospital, 1 Singil-ro, Yeongdeungpo-gu, Seoul 07441, Republic of Korea; hwangjihyo36@gmail.com; 2Incheon Terminal Orthopedics, Inha-ro 489beon-gil, Namdong-gu, Incheon 21574, Republic of Korea; 2wo02wo0@naver.com; 3International Association of Regenerative Medicine, Namdong-gu, Incheon 21574, Republic of Korea; 4MSKUS, San Diego, CA 92084, USA; 5The Board of Clinical Research, The International Association of Musculoskeletal Medicine, Kowloon, Hong Kong; painfreedoc22@gmail.com (T.S.); anwar@ummc.edu.my (A.S.); 6Faculty of Medicine, The University of Hong Kong, Hong Kong; 7Faculty of Medicine, The Chinese University of Hong Kong, New Territory, Hong Kong; 8The Board of Clinical Research, The Hong Kong Institute of Musculoskeletal Medicine, Kowloon, Hong Kong; 9Korea Association of Cyriax Orthopaedic Medicine, Yangpyeong 12522, Republic of Korea; yeskimmy@daum.net (S.K.); chirojk@naver.com (J.K.); 10Rolfing Spine Posture Institute, Seoul 06236, Republic of Korea; pilateshe@naver.com (H.K.); chirodrkang@naver.com (J.K.); 11Department of Sports Rehabilitation, Cheongju University, Cheongju 28503, Republic of Korea; 12Department of Physical Medicine and Rehabilitation, Chi Mei Medical Center, Tainan 710, Taiwan; dr.daniel@gmail.com; 13A Tempo Regeneration Center for Musicians, Tainan 700, Taiwan; 14Department of Physical Medicine and Rehabilitation, Hermina Podomoro Hospital, North Jakarta 14350, Indonesia; 15Department of Physical Medicine and Rehabilitation, Medistra Hospital, South Jakarta 12950, Indonesia; 16Physical Medicine and Rehabilitation, Synergy Clinic, West Jakarta 11510, Indonesia; 17Department of Rehabilitation Medicine, Universiti Malaya, Kuala Lumpur 50603, Malaysia; 18Independent Researcher, Roeland Park, KS 66205, USA; deanreevesmd@gmail.com

**Keywords:** scapula, stress fracture, boxing, ultrasound, computed tomography, kinetic chain, athletic injuries, diagnostic imaging, upper extremity

## Abstract

**Background and Clinical Significance**: Scapular stress fractures are exceptionally rare in athletes and are notoriously difficult to diagnose due to their subtle presentation and poor sensitivity on initial radiographs. This case report describes the diagnostic challenge of a scapular body stress fracture in an elite boxer who initially presented with wrist pain. **Case Presentation**: A 19-year-old right-hand-dominant female elite boxer presented with a three-month history of bilateral wrist pain. Initial examination and MRI were consistent with a triangular fibrocartilage complex (TFCC) injury. Despite conservative management, her symptoms persisted, and she subsequently developed mechanical right shoulder pain and a sensation of instability. Physical examination revealed scapular dyskinesis, with a positive push-up test and weakness on punch protraction. Plain radiographs of the scapula were unremarkable. Point-of-care musculoskeletal ultrasound (MSK US) identified a cortical irregularity at the medial scapular border. A subsequent computed tomography (CT) scan obtained at three-month follow-up definitively confirmed a stress fracture at this site. Treatment focused on scapular stabilization via prolotherapy and activity modification, leading to symptomatic resolution and a successful return to sport. **Conclusions:** This case underscores the importance of evaluating the entire kinetic chain in athletes presenting with focal complaints. It demonstrates the utility of MSK US as an effective initial screening tool for cortical stress fractures and highlights the necessity of CT for definitive confirmation. Clinicians should maintain a high index of suspicion for scapular stress injuries in overhead athletes with unexplained shoulder dysfunction.

## 1. Introduction

Among elite boxers, 36.8% of all injuries are directly linked to the punching motion, with acute and chronic shoulder injuries comprising 9.7% of these cases [1]. Injuries to the shoulder, elbow, and wrist that occur along the kinetic chain of a punch not only lead to a decline in athletic performance but also result in an average training cessation of 14.2 to 20 days, thereby delaying the athlete’s return to competition. This can have a detrimental impact on an athlete’s entire career [2,3].

Common hand and wrist injuries in boxers include fractures of the fifth metacarpal neck or base, commonly known as “Boxer’s fractures.” While these acute injuries are typically well diagnosed and effectively treated [4], shoulder injuries are also frequently reported, as the shoulder is considered one of the most critical areas affected during boxing training. These injuries encompass shoulder subluxation and dislocation, sprains, tendinitis, and chronic impingement syndrome [3,5,6]. However, in contrast to the extensive research on soft tissue shoulder injuries, scapular body stress fractures remain exceptionally rare in boxers. While upper extremity stress fractures in athletes are increasingly documented [7,8,9], the scapular body represents a striking gap in the boxing literature. Our review identified only one reported case in an elite boxer since 1995 [10]. This diagnostic challenge is particularly relevant given the advancing role of point-of-care ultrasound in detecting occult fractures [11,12].

Repetitive, high-velocity punching motions, such as those performed in boxing, can lead to microtrauma and capsular laxity of the shoulder joint [3]. This cumulative damage may result in decreased muscular strength and a limited range of motion, which can contribute to the development of scapular dyskinesis [5]. When evaluating shoulder pain in boxers, clinicians should consider differential diagnoses, including first rib stress fractures, humeral fractures, superior labral anterior–posterior (SLAP) lesions, and rotator cuff tear [13,14,15]. Dysfunction of the serratus anterior muscle—a key stabilizer during punching—and resultant scapular dyskinesis must also be considered [3,16].

Scapular dyskinesis is defined as an abnormal static positioning or dynamic movement of the scapula and is considered a potential underlying cause of shoulder pain and functional impairment [17], with ongoing research refining its clinical relevance [18]. It is well established that scapular dyskinesis, when repetitive in nature, can contribute to the development of impingement syndrome, rotator cuff injuries, clavicular fractures, and labral lesions [3,17,19]. Nevertheless, the specific impact of repetitive punching on scapular structure, including the potential for stress fracture, remains poorly characterized. Currently, most treatment approaches for scapular dyskinesis focus predominantly on soft tissue and ligament injuries, aiming to restore scapular position and mechanics through either conservative or surgical interventions. However, a significant limitation is the inadequate evaluation of scapular fractures themselves [20].

We present a case of a scapular body stress fracture in an elite boxer, initially detected by musculoskeletal ultrasound and confirmed by computed tomography. This case is notable for the patient’s initial presentation with wrist pain, underscoring the critical importance of a comprehensive kinetic chain evaluation.

## 2. Case Report

On October 4, 2024, a 19-year-old right-hand-dominant female elite boxing athlete presented with a three-month history of bilateral wrist pain. The pain was described as a deep, aching discomfort localized to the ulnar side, exacerbated during hook and uppercut punches. Range of motion of both wrists was full but painful at end-range ulnar deviation and supination. The patient denied any specific traumatic event, such as a fall or direct blow to the shoulder or wrist, consistent with an insidious onset from repetitive microtrauma. Initial physical examination revealed positive findings for the ulnar fovea sign, grind test, and ballottement test. The patient reported that her symptoms worsened with increasing training intensity and that she experienced a decline in performance during afternoon training sessions compared to those held in the morning.

At the time of presentation, the patient had previously undergone an MRI at another hospital and was recommended to undergo arthroscopic surgery (A/S operation). On the T2-weighted coronal image of the MRI, high signal intensity was observed in the TFCC region, suggesting a partial tear or degenerative changes (Appendix A). Wrist X-ray performed at our institution showed no specific findings indicative of DRUJ subluxation or perilunar ligament injury (Appendix A). A provisional diagnosis of DRUJ instability with TFCC injury was made, and conservative management—including prolotherapy, taping, and night splinting—was initiated. However, symptoms improved transiently but recurred with resumed training. The transient improvement with initial conservative care was described as a 40–50% reduction in pain, but symptoms returned to baseline upon attempting high-impact bag work.

After two weeks of treatment, the patient began to complain of right shoulder pain on her dominant side, in addition to the persistent wrist pain. She reported a sensation of shoulder dislocation, a clunking instability around the scapular region, and diffuse discomfort in the upper extremity and chest. Mechanical symptoms of the scapula associated with movement became increasingly pronounced. Due to the persistent pain and these mechanical features, scapular dyskinesis was suspected, and a physical examination was conducted. The push-up test and protraction assessed with the boxed punch test were positive. Given the concern for serratus anterior weakness potentially related to long thoracic nerve dysfunction, scapular radiographs and musculoskeletal (MSK) ultrasound were performed.

Radiographs, including true anteroposterior, Y, and axillary views, showed no abnormality [Figure 1], but the MSK ultrasound revealed cortical irregularity in the right scapular body, corresponding to the site of pain. Two distinct longitudinal views were obtained: one directly over the medial scapular border showing a subtle cortical irregularity, and another medial to the border revealing a more extensive cortical disruption with a longer fracture line [Figure 2]. Sonoguided digital palpation was performed, and localized tenderness was observed at the site of cortical irregularity, leading to a provisional diagnosis of scapular fracture. The chronological sequence of clinical events, diagnostic imaging, and interventions is summarized in Table 1. Following the suspicion of a scapular fracture, treatment focused on both potential fracture healing and the restoration of scapular stability [21]. Prolotherapy injections were administered to the right scapular medial border and surrounding ligaments to promote structural stability. Prolotherapy was administered using a solution of 10% dextrose. Under ultrasound guidance, approximately 5 mL of the solution was injected into the periosteum and surrounding tendons at the site of cortical irregularity on the medial scapular border. This procedure was repeated bi-weekly for a total of 4 sessions over two months.

During the initial 4 weeks of treatment, the patient was advised to completely refrain from punching activities but was permitted to maintain cardiovascular fitness and lower body strength training. From weeks 5 to 8, she gradually reintroduced shadow boxing and low-intensity pad work, with a focus on proper scapular mechanics.

Over a two-month follow-up period, the patient showed a gradual increase in punching power, and discomfort in the scapular region improved. At the three-month follow-up, a computed tomography (CT) scan of the right scapula was obtained to confirm the sonographic diagnosis. The CT images clearly revealed a cortical break with associated periosteal reaction at the medial border of the scapular body, corresponding precisely to the site of sonographic irregularity and point tenderness, thus definitively confirming the diagnosis of a stress fracture (Figure 3).

High-intensity sparring and heavy bag training were resumed only after this CT confirmation of fracture healing. As a result, she was able to return to full training.

### Long-Term Follow-Up and Functional Recovery

To assess long-term recovery, the patient returned for a comprehensive medical check-up in August 2025, approximately 10 months post-diagnosis. She reported complete and sustained resolution of both the right wrist and scapular pain, with full return of punching power. Physical examination revealed no tenderness over the medial scapular border, and scapular kinematics were normal with a negative push-up test. She had been participating in full-intensity boxing training without limitations.

This case report is a retrospective study based on anonymized medical records and documents, with no direct contact with the patient or collection of personal identifiable information; therefore, it qualifies for exemption from review by the Institutional Review Board (IRB). Written informed consent for publication of clinical details and treatment outcomes was obtained from the patient. This study was conducted in accordance with the ethical principles of the Declaration of Helsinki.

## 3. Discussion

Upper extremity stress fractures, though uncommon, represent a significant diagnostic consideration in athletes engaging in repetitive overhead or impact activities [9]. In this context, the decision to employ MSK US as the initial advanced imaging modality after negative radiographs was multifactorial. Firstly, US offers the distinct advantage of dynamic, real-time evaluation and allows for direct correlation with point tenderness via sonopalpation [22]. This interactive component is invaluable in localizing occult bony lesions that are not apparent on static images. Secondly, US is highly sensitive in detecting cortical discontinuities and early periosteal reactions, which are the hallmarks of stress fractures [22,23,24]. This is supported by recent evidence confirming the high diagnostic accuracy of ultrasound in fracture detection [11,12], a finding corroborated by case series demonstrating the utility of ultrasound in identifying occult fractures of the extremities and tibia that were not apparent on initial radiographs [25,26]. While MRI is an excellent modality for assessing bone marrow edema and associated soft tissue pathology, CT remains the gold standard for characterizing cortical integrity [10]. In our clinical context, US served as a rapid, cost-effective, and accessible point-of-care tool to triage the patient and justify the subsequent use of a more specialized and higher-radiation-dose modality like CT. This diagnostic pathway—US for initial high-sensitivity screening followed by CT for definitive confirmation—represents a pragmatic and efficient algorithm in the sports medicine setting.

This case is the first reported study to diagnose a scapular body stress fracture using ultrasound that was subsequently confirmed by CT. Research on shoulder injuries in boxing athletes remains limited and has primarily focused on acute injuries diagnosed through MRI. Furthermore, this study is the first to identify and treat an associated shoulder injury through a biomechanical approach in a patient who initially presented with wrist pain rather than shoulder pain. Cortical irregularity and disruption of continuity in the scapula were initially visualized with ultrasound, and the diagnostic accuracy was enhanced through sonoguided digital palpation prior to definitive CT confirmation.

According to previous literature, scapular stress fractures have primarily been reported in athletes participating in various overhead sports such as baseball [27,28], cricket [29], swimming [30] and water polo [31]. However, cases of scapular body stress fractures occurring in boxing athletes are scarcely reported. Most scapular stress fractures reported in the literature involve the acromion or are associated with prosthetic implants and specific risk factors [32,33], making our case of a scapular body fracture in a boxer resulting from repetitive biomechanical overload particularly unusual. Furthermore, imaging modalities commonly used to diagnose similar injuries in existing studies have included X-rays, which have low initial sensitivity, and costly magnetic resonance imaging (MRI) [10,34]. To date, there have been no reports diagnosing scapular body stress fractures using ultrasound or describing the underlying mechanisms. In a case reported by Wyrsch et al. [10], a professional boxer experienced shoulder pain during punching motions; initial X-rays showed no abnormalities, and the scapular stress fracture was only confirmed through subsequent MRI.

Musculoskeletal (MSK) ultrasound is a modality that provides direct visualization of bones, ligaments, tendons, muscles, nerves, and vessels without exposing patients to radiation, and it is widely used in the diagnosis and management of musculoskeletal disorders [35]. In particular, for fracture diagnosis at distal bones, MSK ultrasound has demonstrated sensitivity and specificity comparable to, or even exceeding, that of MRI, making it a valuable diagnostic tool [22]. However, the use of ultrasound in the diagnosis and treatment of shoulder injuries is typically limited to rotator cuff tears and dislocations. Therapeutic applications mainly include intra-articular injections, subacromial space injections, and nerve blocks, while its use for fracture assessment remains limited [23,36,37]. In the present case, musculoskeletal ultrasound served as a highly effective initial screening tool, correctly identifying the cortical abnormality that was occult on plain radiographs. This finding was subsequently confirmed by CT imaging, which is considered the gold standard for characterizing cortical bony pathology. This diagnostic pathway—employing ultrasound for initial point-of-care evaluation followed by CT for definitive confirmation—provides a viable and efficient algorithm for evaluating suspected stress fractures in athletes.

Scapular stress fractures are rare injuries that are prone to delayed diagnosis, as lesions are often not visible on initial plain radiographs, resulting in low diagnostic sensitivity [27,38]. Moreover, the subsequent periosteal reaction may be mistaken for malignant conditions such as osteosarcoma or osteomyelitis, potentially leading to unnecessary invasive procedures such as biopsies [34].

The biomechanics of punching provide a plausible explanation for this injury. The serratus anterior muscle, a primary scapular protractor and stabilizer, exerts significant tensile force on its insertion site along the medial scapular border during the follow-through phase of a punch [39,40]. It is plausible that pre-existing kinetic chain dysfunction from the painful wrist (TFCC injury) led the athlete to subconsciously alter her punching mechanics, leading to compensatory overload and inefficient force dissipation through the serratus anterior [41]. This created a cycle of repetitive microtrauma at the muscle-bone interface, ultimately exceeding the scapula’s remodeling capacity and resulting in a stress fracture [42,43,44]. The observed scapular dyskinesis and serratus weakness on examination further support this mechanism of injury [45].Anatomically, the shoulder and elbow serve as structures that position the hand optimally [46]. Therefore, while it is important to focus on local lesions when addressing wrist pain, an approach considering the entire upper limb kinetic chain and the transmission of biomechanical forces is essential. The punching motion follows a sequential biomechanical process in which force generated at the shoulder is rapidly transmitted through elbow extension and ultimately delivered as rotational power at the forearm.34 Thus, although local pathology causing wrist pain requires attention, it is crucial to adopt a comprehensive approach that accounts for biomechanical force transmission throughout the kinetic chain. The chronology of this case presents a diagnostic dilemma; it remains challenging to determine whether the scapular stress fracture was the primary instigator of kinetic chain dysfunction that manifested as wrist pain, or whether the initial wrist pathology (TFCC injury) led to altered punching mechanics that secondarily induced excessive stress on the scapula.

## 4. Conclusions

This case highlights the importance of considering scapular stress fracture in the evaluation of athletes presenting with shoulder pain or discomfort. Even in the absence of trauma and when the primary symptoms are wrist or hand pain, the possibility of scapular pathology should not be excluded given the biomechanical and kinetic chain relationships of the upper limb. This case demonstrates that point-of-care musculoskeletal ultrasound is a valuable tool for clinicians to identify characteristic features of stress fractures, such as cortical discontinuity and periosteal reaction, which may be occult on initial radiographs. In this clinical scenario, the sequential use of ultrasound for initial screening, followed by confirmatory CT imaging, facilitated a timely diagnosis and guided appropriate conservative management, ultimately enabling a successful return to sport. This pragmatic diagnostic pathway may be particularly useful in the sports medicine setting where rapid, accessible assessment is required.

## Figures and Tables

**Figure 1 diagnostics-15-02565-f001:**
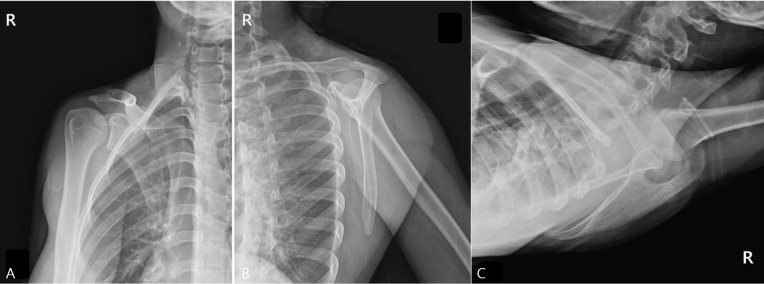
Plain Radiographs of the Right Scapula. These radiographs were taken due to the worsening of wrist symptoms, accompanied by a feeling of instability in the scapular region. (**A**) True anteroposterior (AP) view of the shoulder, (**B**) scapular Y-view, and (**C**) axillary view. All views showed no evidence of a definite fracture line, dislocation, or other specific abnormalities.

**Figure 2 diagnostics-15-02565-f002:**
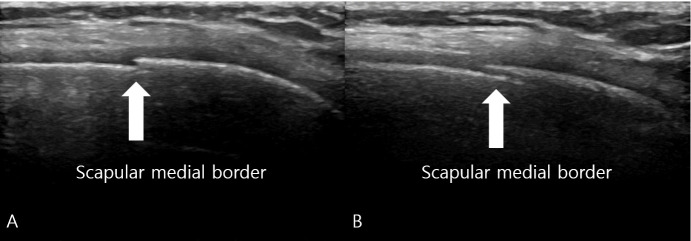
Musculoskeletal Ultrasound of the Right Scapula. (**A**) Longitudinal view obtained directly over the medial border of the scapula reveals a subtle cortical irregularity (white arrow) indicating the fracture site, which was not visible on plain radiographs. (**B**) Longitudinal view obtained medial to the scapular border demonstrates a more extensive cortical disruption with a longer fracture line (white arrow), confirming the location of point tenderness and providing additional evidence of the stress fracture. The sonographic finding of cortical disruption, combined with focal tenderness on sonopalpation, led to a provisional diagnosis of a scapular stress fracture.

**Figure 3 diagnostics-15-02565-f003:**
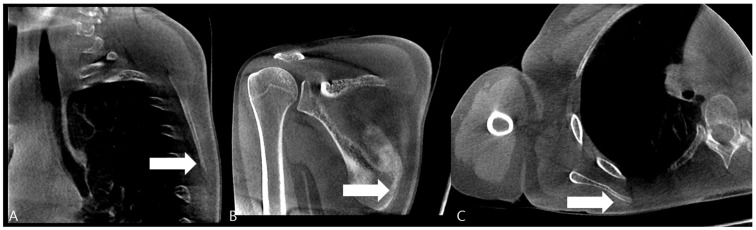
Computed Tomography (CT) of the Right Scapula. (**A**) Cortical breakage and callus formation are observed on the medial side of the scapular in the sagittal image (white arrow). (**B**) Callus formation (white arrow) and periosteal reaction are observed on the inferior part of the scapular medial border in the coronal image. (**C**) Callus formation is observed on the scapula medial border in the axial image (white arrow).

**Table 1 diagnostics-15-02565-t001:** Timeline of Clinical Presentation, Diagnosis, and Management.

Date	Clinical Event/Intervention	Key Findings/Outcomes
July 2024	Onset of bilateral wrist pain	Pain worsened with training intensity
4 October 2024	Initial presentation to clinic	Positive ulnar fovea sign, grind test; provisional diagnosis of DRUJ instability with TFCC injury
Mid-October 2024	Initiation of conservative management for wrist	Prolotherapy, taping, night splinting; transient improvement 40 50% pain reduction)
Late October 2024	Development of right shoulder pain	Sensation of instability, clunking around scapula; scapular dyskinesis suspected
Late October 2024	Scapular imaging performed	X-rays normal; MSK ultrasound revealed cortical irregularity at medial scapular border; provisional diagnosis of scapular stress fracture
November–December 2024	Prolotherapy treatment phase	Four sessions of ultrasound-guided prolotherapy 10% dextrose, bi-weekly) to scapular medial border
November 2024 January 2025	Graded return to activity	Initial 4 weeks refrain from punching; weeks 5 8 gradual reintroduction of shadow boxing and low-intensity pad work
January 2025	3-month follow-up; CT scan obtained	CT confirmed stress fracture with callus formation at medial scapular border
February 2025	Full return to training	High-intensity sparring and heavy bag training resumed following CT confirmation of healing
August 2025	Long-term follow-up (medical check-up)	Complete resolution of symptoms; full return of punching power; participation in full-intensity training without limitations

## Data Availability

Data related to this study has been included in the manuscript.

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
