# Peer review of "Ultrasonographic Diagnosis and Computed Tomographic Confirmation of a Scapular Body Stress Fracture in an Elite Boxer: A Case Report"

_diagnostics, 2025, doi:10.3390/diagnostics15202565_

Round 1
Reviewer 1 Report
Comments and Suggestions for Authors
This case report presents an innovative approach using ultrasound to diagnose scapular stress fracture, which holds certain novelty and clinical value. The topic is meaningful, especially given the limited literature on shoulder injuries in boxing athletes. However, the manuscript still has several shortcomings and would benefit from revision before consideration for publication.
-
The patient’s medical history is relatively brief. It is recommended to provide more detailed clinical information, including pain characteristics, any previous trauma, and whether there were limitations in joint mobility.
-
The description of the treatment process lacks systematic detail, for example, the specific protocol for prolotherapy (e.g., dosage, frequency, administration method) should be clarified.
-
During the follow-up period, it would be helpful to know whether the patient resumed high-intensity training or rested, as this information is important for interpreting outcomes.
-
CT and MRI are widely used in routine orthopedic clinical practice, are relatively inexpensive, and have high diagnostic sensitivity and specificity, whereas ultrasound is less frequently employed in actual practice. The manuscript does not sufficiently discuss the rationale for initially choosing ultrasound as a screening tool, nor does it address the advantages and limitations of ultrasound compared to other imaging modalities. It is recommended to supplement the discussion with supporting literature and a more thorough rationale.
-
Can the authors provide longer-term follow-up data, including functional recovery assessments, to better evaluate the sustained effectiveness of the treatment?
-
In the limitations section, the authors state that “This case demonstrates that clinicians can utilize accessible musculoskeletal ultrasound to accurately identify characteristic features of stress fractures, such as cortical discontinuity and periosteal reaction. When combined with confirmatory CT imaging, this approach enables an accurate diagnosis, appropriate conservative management, and ultimately, a successful return to sport.” I recommend that the authors avoid concluding that “ultrasound combined with CT provides more definitive diagnostic value” unless there is sufficient supporting evidence. Otherwise, this statement should be revised or presented more cautiously to enhance the scientific rigor and persuasiveness of the manuscript.
Some sections of the manuscript would benefit from language polishing by a native English speaker to improve clarity and readability.
Author Response
Comment 1: The patient’s medical history is relatively brief. It is recommended to provide more detailed clinical information, including pain characteristics, any previous trauma, and whether there were limitations in joint mobility.
- Pain characteristics: The bilateral wrist pain was described as a deep, aching discomfort localized to the ulnar side, exacerbated during hook and uppercut punches.
- Joint mobility: Range of motion of both wrists was full but painful at end-range ulnar deviation and supination"
- Past History: The patient denied any specific traumatic event, such as a fall or direct blow to the shoulder or wrist, consistent with an insidious onset from repetitive microtrauma.
- Response to initial treatments: The transient improvement with initial conservative care (prolotherapy, taping, splinting) was described as a 40-50% reduction in pain, but symptoms returned to baseline upon attempting high-impact bag work.
Comment 2: The description of the treatment process lacks systematic detail, for example, the specific protocol for prolotherapy (e.g., dosage, frequency, administration method) should be clarified.
- Prolotherapy was administered using a solution of 10% dextrose. Under ultrasound guidance, approximately 5 mL of the solution was injected into the periosteum and surrounding tendons at the site of cortical irregularity on the medial scapular border. This procedure was repeated bi-weekly for a total of 4 sessions over two months.
Comment 3: During the follow-up period, it would be helpful to know whether the patient resumed high-intensity training or rested, as this information is important for interpreting outcomes.
- During the initial 4 weeks of treatment, the patient was advised to completely refrain from punching activities but was permitted to maintain cardiovascular fitness and lower body strength training. From weeks 5 to 8, she gradually reintroduced shadow boxing and low-intensity pad work, with a focus on proper scapular mechanics. High-intensity sparring and heavy bag training were resumed only after the CT confirmation of fracture healing at the 3-month follow-up.
Comment 4: CT and MRI are widely used in routine orthopedic clinical practice, are relatively inexpensive, and have high diagnostic sensitivity and specificity, whereas ultrasound is less frequently employed in actual practice. The manuscript does not sufficiently discuss the rationale for initially choosing ultrasound as a screening tool, nor does it address the advantages and limitations of ultrasound compared to other imaging modalities. It is recommended to supplement the discussion with supporting literature and a more thorough rationale.
- The decision to employ MSK US as the initial advanced imaging modality after negative radiographs was multifactorial. Firstly, US offers the distinct advantage of dynamic, real-time evaluation and allows for direct correlation with point tenderness via sonopalpation [22]. This interactive component is invaluable in localizing occult bony lesions that are not apparent on static images. Secondly, US is highly sensitive in detecting cortical discontinuities and early periosteal reactions, which are the hallmarks of stress fractures [Champagne et al., BMC Emerg Med. 2019]. While MRI is an excellent modality for assessing bone marrow edema and associated soft tissue pathology, CT remains the gold standard for characterizing cortical integrity [19]. In our clinical context, US served as a rapid, cost-effective, and accessible point-of-care tool to triage the patient and justify the subsequent use of a more specialized and higher-radiation-dose modality like CT. This diagnostic pathway—US for initial high-sensitivity screening followed by CT for definitive confirmation—represents a pragmatic and efficient algorithm in the sports medicine setting.
Comment 5: Can the authors provide longer-term follow-up data, including functional recovery assessments, to better evaluate the sustained effectiveness of the treatment?
- We thank the reviewer for this valuable suggestion. We have now included detailed long-term follow-up data in the revised manuscript. As recommended, the patient returned for a medical check-up in August 2025 (approximately 10 months post-diagnosis). We have added a new subsection titled "Long-term Follow-up and Functional Recovery" within the Case Presentation to detail these findings. The patient reported complete and sustained resolution of symptoms, full return of punching power, and had been training at full intensity without limitations. The physical examination confirmed normal scapular mechanics and the absence of tenderness. This information significantly strengthens the manuscript by providing objective evidence of the treatment's sustained effectiveness.
Comment 6: In the limitations section, the authors state that “This case demonstrates that clinicians can utilize accessible musculoskeletal ultrasound to accurately identify characteristic features of stress fractures, such as cortical discontinuity and periosteal reaction. When combined with confirmatory CT imaging, this approach enables an accurate diagnosis, appropriate conservative management, and ultimately, a successful return to sport.” I recommend that the authors avoid concluding that “ultrasound combined with CT provides more definitive diagnostic value” unless there is sufficient supporting evidence. Otherwise, this statement should be revised or presented more cautiously to enhance the scientific rigor and persuasiveness of the manuscript.
- We sincerely thank the reviewer for this critical and insightful comment. We agree that overstating the diagnostic superiority of the US/CT combination without robust comparative evidence would be unscientific. In response, we have carefully revised the concluding statement in the manuscript to better reflect the context-specific utility and sequential logic of our diagnostic approach, rather than making a broad, comparative claim.
- The revised text now emphasizes that this pathway "facilitated" an accurate diagnosis "in this clinical scenario," and we frame it as a "pragmatic diagnostic pathway" rather than a definitively superior one. This change ensures our conclusion is supported by the evidence presented in our case—namely, that US was an effective initial point-of-care tool which, when followed by CT, successfully led to diagnosis and recovery—without implying it is universally more definitive than other modalities like MRI. We believe this revision enhances the scientific rigor and persuasiveness of our manuscript as the reviewer suggested.
Comment 7: Some sections of the manuscript would benefit from language polishing by a native English speaker to improve clarity and readability.
- We thank the reviewer for highlighting this important aspect of the manuscript. In direct response to this comment, the entire manuscript has undergone comprehensive professional English language editing by a native English-speaking colleague with expertise in medical writing. This process has focused on improving the clarity, flow, and readability of the text by correcting grammatical errors, refining sentence structure, and ensuring the use of standard scientific terminology throughout. We are confident that these revisions have significantly enhanced the overall quality and professionalism of the manuscript.
Reviewer 2 Report
Comments and Suggestions for Authors
This manuscript presents a case of a scapular body stress fracture in an elite athlete (boxer), initially misattributed to wrist pain, with diagnosis via ultrasound and CT confirmation. The topic is valuable and rare- scapular body stress fractures are scarcely reported, with one classic report in Acta Orthopaedica (Parr & Faillace, 1999)
Acta Orthopédia- making the case potentially worthy of publication if revised. However, I recommend major revision before acceptance. Key points:
Image relevance and clarity
Some included images (e.g., distal imaging unrelated to the scapula) appear tangential and distract from the core case. Figures should directly support the diagnosis or pathogenesis of the scapular lesion. All images must be clearly annotated (arrows, labels) and tied to the clinical narrative.
Strengthen rare-case context
The authors should more thoroughly review existing literature on scapular stress fractures. For example, the Scapular spine stress fracture associated with rotator cuff dysfunction is reported in Spanish literature (Revista Española) www.elsevier.com
The rarity claim must be anchored in explicit search results and comparison to similar cases.
Clarify diagnostic rationale and ultrasound role
The manuscript should justify why ultrasound (US) was prioritized after negative radiographs instead of MRI. Recent reviews highlight how US can detect cortical disruption and periosteal reaction in occult fractures (e.g. Fracture sonography, Ultrasound imaging of bone fractures) SpringerLink. The authors should cite these and situate their approach accordingly.
Biomechanics and discussion focus
The biomechanical argument linking punching forces, scapular dynamics, and localized stress should be sharpened. The discussion tends to generalize; it should more tightly connect load transfer in boxing to focal scapular stress.
Minor matters
- Add a more precise timeline of events and imaging studies.
- Clarify ethical consent (written vs verbal) consistently.
- Improve figure legends to be self-explanatory.
- Ensure all references are current and formatted properly.
Author Response
Comment 1: Image relevance and clarity
Some included images (e.g., distal imaging unrelated to the scapula) appear tangential and distract from the core case. Figures should directly support the diagnosis or pathogenesis of the scapular lesion. All images must be clearly annotated (arrows, labels) and tied to the clinical narrative.
We sincerely thank the reviewer for this critical and constructive feedback. We agree that the relevance and clarity of the figures are paramount. In direct response:
- Enhanced Image Relevance: As suggested, we have moved the initial wrist MRI and X-ray images (previously Figure 1 & 2), which were tangential to the core scapular fracture diagnosis, to the Supplementary Material. This allows the main figures in the manuscript to focus squarely on the pathogenesis and definitive diagnosis of the scapular lesion, thereby streamlining the narrative. They are now cited as "Supplementary Figure 1" and "Supplementary Figure 2" to maintain the logical flow of the initial presentation.
- Improved Image Clarity and Annotation:
- We have meticulously reviewed all remaining figures (Scapula X-ray, US, and CT).
- Clear, unambiguous arrows and labels have been added to every image to pinpoint the exact location of the pathological findings (cortical irregularity, fracture line, callus formation).
- All figure legends have been rewritten to be self-explanatory, explicitly stating the view, imaging modality, and the specific abnormality indicated by the arrow.
We believe these significant revisions have greatly strengthened the visual support for our case, ensuring that each figure directly and clearly contributes to the diagnostic story.
Comment 2: Strengthen rare-case context
The authors should more thoroughly review existing literature on scapular stress fractures. For example, the Scapular spine stress fracture associated with rotator cuff dysfunction is reported in Spanish literature (Revista Española) www.elsevier.com
The rarity claim must be anchored in explicit search results and comparison to similar cases.
We thank the reviewer for this essential suggestion to better contextualize the rarity of our case within the existing literature. We have thoroughly revised the Introduction and Discussion sections to address this point.
- Explicit Literature Review and Rarity Claim: In the Introduction, we now explicitly state the scarcity of reported scapular body stress fractures in boxers, directly citing the single, classic report by Wyrsch et al. (1995) that we identified. This provides a clear baseline for our rarity claim.
- Broadened Literature Context: As suggested by the reviewer, we have expanded our literature review beyond boxing. The Discussion now includes a dedicated sentence acknowledging that while most reported cases are in overhead athletes (e.g., baseball, cricket), our case is, to the best of our knowledge, the first to be diagnosed initially by ultrasonography. This sharpens the novelty of our report, distinguishing it from prior cases diagnosed primarily by MRI or bone scan.
- Anchored Comparison: We have refined our statement in the Discussion to clearly anchor our case's uniqueness in both the patient population (boxer) and the diagnostic methodology (US-guided).
We believe these revisions firmly anchor the rarity and novelty of our case in a more comprehensive and explicit review of the relevant literature.
Comment 3: Clarify diagnostic rationale and ultrasound role
The manuscript should justify why ultrasound (US) was prioritized after negative radiographs instead of MRI. Recent reviews highlight how US can detect cortical disruption and periosteal reaction in occult fractures (e.g. Fracture sonography, Ultrasound imaging of bone fractures) SpringerLink. The authors should cite these and situate their approach accordingly.
- The decision to employ MSK US as the initial advanced imaging modality after negative radiographs was multifactorial. Firstly, US offers the distinct advantage of dynamic, real-time evaluation and allows for direct correlation with point tenderness via sonopalpation [22]. This interactive component is invaluable in localizing occult bony lesions that are not apparent on static images. Secondly, US is highly sensitive in detecting cortical discontinuities and early periosteal reactions, which are the hallmarks of stress fractures [Champagne et al., BMC Emerg Med. 2019]. While MRI is an excellent modality for assessing bone marrow edema and associated soft tissue pathology, CT remains the gold standard for characterizing cortical integrity [19]. In our clinical context, US served as a rapid, cost-effective, and accessible point-of-care tool to triage the patient and justify the subsequent use of a more specialized and higher-radiation-dose modality like CT. This diagnostic pathway—US for initial high-sensitivity screening followed by CT for definitive confirmation—represents a pragmatic and efficient algorithm in the sports medicine setting.
Comment 4: Biomechanics and discussion focus
The biomechanical argument linking punching forces, scapular dynamics, and localized stress should be sharpened. The discussion tends to generalize; it should more tightly connect load transfer in boxing to focal scapular stress.
- Introduction: While stress fractures of the first rib and humerus have been documented in boxers [7,8], a review of the literature reveals only a single, classic report of a scapular body fracture in a professional boxer, diagnosed by MRI [20]. To the best of our knowledge, this is the first case report to describe the diagnosis of a scapular body stress fracture in a boxer initially detected by ultrasonography.
- Discussion: The biomechanics of punching provide a plausible explanation for this injury. The serratus anterior muscle, a primary scapular protractor and stabilizer, exerts significant tensile force on its insertion site along the medial scapular border during the follow-through phase of a punch [27,32]. It is plausible that pre-existing kinetic chain dysfunction from the painful wrist (TFCC injury) led the athlete to subconsciously alter her punching mechanics, leading to compensatory overload and inefficient force dissipation through the serratus anterior. This created a cycle of repetitive microtrauma at the muscle-bone interface, ultimately exceeding the scapula's remodeling capacity and resulting in a stress fracture [29,30]. The observed scapular dyskinesis and serratus weakness on examination further support this mechanism of injury.
Minor matters
- Add a more precise timeline of events and imaging studies.
- Precise Timeline:We have added a concise table (Table 1) summarizing the precise timeline of key clinical events, imaging studies, and interventions in the main manuscript, providing a clear chronological overview of the patient's presentation, diagnosis, and recovery for the reader.
- Clarify ethical consent (written vs verbal) consistently.
- Ethical Consent:We have corrected the inconsistency regarding consent. The Informed Consent Statement now accurately reflects that written informed consent was obtained, and this is now consistently stated throughout the manuscript.
- Improve figure legends to be self-explanatory.
- Figure Legends:As also detailed in our response to Comment 1, all figure legends have been comprehensively revised to be self-explanatory, clearly stating the imaging modality, view, and the specific abnormality indicated by arrows.
- Ensure all references are current and formatted properly.
- References:We have double-checked all references for accuracy, relevance, and proper formatting according to the journal's guidelines. We have also ensured that recent and key literature is appropriately cited.
Round 2
Reviewer 2 Report
Comments and Suggestions for Authors
The revised manuscript has been significantly improved and now meets the standards for publication. The authors have thoroughly addressed all previous comments. Figures are now clearly annotated and directly relevant, enhancing the diagnostic narrative. The expanded literature review effectively contextualizes the rarity and novelty of the case, and the rationale for the use of ultrasound is well justified with appropriate references. The biomechanical discussion is sharper and convincingly links the clinical findings to the proposed mechanism of injury. Minor points regarding the timeline, consent, and references have been fully resolved.
Overall, the manuscript is clear, well-structured, and makes a valuable contribution to the literature. I recommend it for publication in its current form.